# Chromosome-Level Genome Assembly and Annotation of the Highly Heterozygous *Phallus echinovolvatus* Provide New Insights into Its Genetics

**DOI:** 10.3390/jof11010062

**Published:** 2025-01-15

**Authors:** Mengya An, Ruoxi Liang, Yanliu Chen, Jinhua Zhang, Xiuqing Wang, Xing Li, Guohua Qu, Junfeng Liang

**Affiliations:** 1Research Institute of Tropical Forestry, Chinese Academy of Forestry, Guangzhou 510520, China; anmy0505@163.com (M.A.); xiaoliuchen0316@126.com (Y.C.); jhzhang0111@163.com (J.Z.); clearmoon2008@126.com (X.W.); lixing@scbg.ac.cn (X.L.); qgh0107@163.com (G.Q.); 2College of Forestry, Nanjing Forestry University, Nanjing 210037, China; 3Honors College, Northwestern Polytechnical University, Xi’an 710129, China; liangruoxi@mail.nwpu.edu.cn

**Keywords:** *Phallus echinovolvatus*, whole genome, dikaryon, comparative genomics, CAZymes, secondary metabolism, terpene

## Abstract

*Phallus echinovolvatus* is a well-known edible and medicinal fungus with significant economic value. However, the available whole-genome information is lacking for this species. The chromosome-scale reference genome (Monop) and two haploid genomes (Hap1 and Hap2) of *P. echinovolvatus*, each assembled into 11 pseudochromosomes, were constructed using Illumina, PacBio-HiFi long-read sequencing, and Hi-C technology. The Monop had a size of 36.54 Mb, with 10,251 predicted protein-coding genes and including 433 carbohydrate-active enzyme genes, 385 cytochrome P450 enzyme genes, and 42 gene clusters related to secondary metabolite synthesis. Phylogenetic and collinearity analysis revealed a close evolutionary relationship between *P. echinovolvatus* and *Clathrus columnatus* in the core Phallales clade. Hap1 and Hap2 had sizes of 35.46 Mb and 36.11 Mb, respectively. Collinear relationships were not observed for 15.38% of the genes in the two haplotypes. Hap1 had 256 unique genes, and Hap2 had 370 unique genes. Our analysis of the *P. echinovolvatus* genome provides insights into the genetic basis of the mechanisms underlying the metabolic effects of bioactive substances and will aid ongoing breeding efforts and studies of genetic mechanisms.

## 1. Introduction

*Phallus echinovolvatus* (M. Zang, D.R. Zheng & Z.X. Hu) Kreisel, originally discovered in Hunan Province, China, in 1988, was initially named *Dictyophora echinovolvata* M. Zang, D.R. Zheng & Z.X. Hu [1]. This fungus is widely distributed in regions south of the Yangtze River in China and in Southeast Asia [2,3]. The genus *Dictyophora* was initially established to classify a group of fungi within the family Phallaceae, characterized by a distinctive net-like “skirt” hanging from the cap. In the early 19th century, these fungi were grouped under the genus *Dictyophora* Desvaux. However, in 1996, Kreisel reduced *Dictyophora* as a section within the genus *Phallus* (P. sect. *Dictyophora* (Desvaux) Kreisel) [4], and the species name was revised to *Phallus echinovolvatus* (M. Zang, D.R. Zheng & Z.X. Hu) Kreisel. Recent molecular phylogenetic studies have shown that the presence or absence of the “skirt” is insufficient to justify the independent taxonomic status of *Dictyophora* [5,6]. These findings provide strong evidence for the taxonomic adjustment, which has since been widely accepted by mycologists [7,8]. Nevertheless, some studies continue to use the genus name *Dictyophora*.

*P. echinovolvatus* is rich in various nutrients and is widely consumed as a functional food in daily life in countries such as China, Japan, Germany, and North America [9,10]. It also has significant biomedical effects. Previous studies have shown that it plays an important role in the immune system, including in tumor cell inhibition and antibiosis [11,12,13,14]. The genome is essential for molecular and genetic research in macrofungi. Few genomic and genetic studies of *P. echinovolvatus* have been conducted compared with other edible mushrooms, such as *Lentinula edodes* [15], *Flammulina velutipes* [16], *Auricularia heimuer* [17], *Morchella importuna* [18], and *Agaricus bisporus* [19]. This research gap has greatly impeded downstream investigations and the utilization of this mushroom. Although the genomes of *Phallus indusiatus* [20] and *Phallus rubrovolvatus* [21,22] have been published, the genome of *P. echinovolvatus* has not yet been sequenced. The molecular basis and evolution of the component biosynthesis in *P. echinovolvatus* are rarely reported due to the lack of a high-quality reference genome.

Microscopic observations have revealed distinctive clamp connections, indicating that the strain (RITF7875) is a dikaryon (Figure 1a). In macrofungi, the presence of two haploid nuclei within a single cell is a widespread phenomenon. This specialized dikaryotic structure poses major challenges to the assembly of macrofungal genomes. Following the emergence of high-fidelity (HiFi) and chromosome conformation capture (Hi-C) sequencing technologies, assembly tools designed for HiFi data, such as Hifiasm and HiCanu, have been used for the assembly of the two haploid genomes of heterozygous organisms [23]. The use of HiFi and Hi-C sequencing strategies has facilitated the phasing and assembly of chromosome-level genomes for several eukaryotic organisms, such as humans [24], *Takifugu ocellatus* [25], diploid *Suaeda glauca* [26], walnut [27], apple [28], *Puccinia triticina* [29], and *Puccinia polysora* [30].

We generated a chromosome-level reference genome and two haploid genomes of *P. echinovolvatus* by integrating PacBio, Illumina, and Hi-C sequencing data. The total length of the reference genome was 43.85 Mb, with a contig N50 of 1.20 Mb. The assembled sequences were anchored to 11 pseudochromosomes with an integration efficiency of 83.58%. The total lengths of Hap1 and Hap2 were 37.68 Mb and 38.46 Mb, with contig N50 lengths of 0.65 Mb and 0.73 Mb, respectively. Approximately 35.46 Mb (94.16%) of Hap1 and 36.11 Mb (93.98%) of Hap2 were assigned to 11 pseudochromosomes. A total of 10,251, 9316, and 9328 protein-coding gene models were predicted for the Monop, Hap1, and Hap2, respectively. In summary, we present novel genomic information about *P. echinovolvatus* through gene annotation and comparative genomic analysis. The genome serves as a direct and comprehensive representation of a species’ genetic information, offering insights unclouded by the effects of convergent evolution or morphological plasticity. This clarity makes genomic analysis an invaluable tool for exploring evolutionary relationships, genetic diversity, and functional traits, bypassing the confounding influences of superficial similarities or adaptive morphological changes. The genetic data generated in this study will significantly contribute to the study of the taxonomy and evolutionary relationships within the Phallaceae family, providing a crucial theoretical foundation for future genetic breeding and the development of active compounds in *P. echinovolvatus*.

## 2. Materials and Methods

### 2.1. Fungal Strain and Genome Sequencing

The dikaryotic *P. echinovolvatus* strain (RITF7875) used in this study was derived from the Forest Resources and Protection Laboratory of the Research Institute of Tropical Forestry, Chinese Academy of Forestry (RITF) (Figure 1). The isolated and purified vegetative mycelia were cultured in a liquid PD medium (30% potato, 2% glucose) for 14 days. Mycelium collected through centrifugation was flash-frozen in liquid nitrogen and stored in a refrigerator at −80 °C.

The high-quality genomic DNA from *P. echinovolvatus* (RITF7875) was extracted using an Omega Bio-Tek Fungal DNA extraction kit (E.Z.N.A.^®^ Fungal DNA Kit, Omega Bio-Tek, Norcross, GA, USA) according to the manufacturer’s instructions. The library was constructed using the Illumina TruSeq^TM^ Nano DNA Sample Prep Kit (Illumina, San Diego, CA, USA) method (with an insert size of 450 bp). After library construction, paired-end sequencing was performed using the Illumina NovaSeq 6000 platform (Shanghai BIOZERON Co., Ltd., Shanghai, China). To enhance the accuracy of the subsequent assembly, adapter sequences were removed from the raw data using Trimmomatic v0.39 software [31] after quality control by FastQC v0.11.9 [32].

The high-molecular-weight genomic DNA was then sheared to a target size of 15–20 kb, and a SMRTbell library was constructed. The genomic library was sequenced in circular consensus sequencing (CCS) mode on the PacBio Sequel II platform. To obtain a more accurate assembly, the original sequencing data were processed as follows: (1) polymerase reads with a length of less than 200 bp were removed; (2) polymerase reads with a quality score of less than 0.80 were removed; (3) CCS reads were extracted from polymerase reads, and adapter sequences were removed; and (4) CCS reads with a length of less than 200 bp were removed.

For Hi-C sequencing, cells were treated with formaldehyde to crosslink the DNA with the proteins, which preserves the conformation of the DNA. Following cell lysis, the crosslinked DNA was treated with a restriction enzyme to generate sticky ends. After end-polishing and repair, biotin was introduced to label the oligonucleotide ends. Adjacent DNA fragments were ligated using DNA ligase. Protein digestion was performed to release the DNA from the crosslinked state, followed by DNA purification and fragmentation into 500–700 bp fragments. The biotin-labeled DNA was captured using streptavidin magnetic beads for library construction. After library preparation, Hi-C sequencing was performed using the Illumina NovaSeq 6000 platform.

### 2.2. Genome Assembly and Assessment

Before assembly, k-mer analysis was used to estimate various genome characteristics. The Illumina sequencing data were analyzed using 21-mers to estimate the genome size, heterozygosity, and repeatability.

The software Hifiasm (https://github.com/chhylp123/hifiasm, accessed on 5 November 2023) was used to assemble the HiFi reads [33]. The ALLHIC tool (https://github.com/tanghaibao/allhic, accessed on 10 November 2023) was used to link the genome contigs or scaffolds [34]. Hi-C-assisted assembly is based on the principle that cis-interactions are stronger than trans-interactions, and the strength of cis-interactions increases at closer distances. Therefore, the contigs or scaffolds were clustered, sorted, and oriented to construct the Hi-C-based chromosome-level assembly, followed by heatmap visualization to display the chromosome contacts [35,36]. BUSCO v4.1.2 [37] was used to assess the completeness and accuracy of the *P. echinovolvatus* genome.

### 2.3. Component Prediction

We used AUGUSTUS v3.2.3 [38] for the de novo gene prediction, and the alignments were performed using GeneWise v2.4 [39] to identify gene-coding regions and intron regions. Subsequently, EVidenceModeler v1.1.1 [40] software was used to integrate the aforementioned results and predict all the protein-coding genes. RepeatMask (http://www.repeatmasker.org, accessed on 5 January 2024) [41] was used to identify interspersed repeats (IRs) by aligning the sequences against a known repetitive sequence database. TRNAscan-SE v2.0.7 [42] was used to predict the transfer RNA (tRNA) regions and the secondary structures of tRNAs. Ribosomal RNA (rRNA) predictions were conducted using rRNAmmer v1.2 [43]. The Rfam v13.0 [44] was used to make comparisons against the RNA families (Rfam)database and acquire annotations, and its integrated search tool (with the default parameters) was used to identify small RNAs (sRNAs), small nuclear RNAs (snRNAs), and microRNAs (miRNAs).

### 2.4. Gene Annotation

The predicted genes were subjected to BLAST alignment against the Non-Redundant (NR), the Kyoto Encyclopedia of Genes and Genomes (KEGG), the Evolutionary Genealogy of Genes: Non-supervised Orthologous Groups (eggNOG), the Gene Ontology (GO), and the Swiss-Prot databases to predict gene functions. PfamScan v1.6, provided by the Pfam database, was used to identify structural domains. TMHMM v2.0 was used to predict transmembrane domain structures. To classify members of the cytochrome P450 gene family in *P. echinovolvatus*, BLAST v 2.10.1 was used to align all the protein sequences of *P. echinovolvatus* against the Fungal Cytochrome P450 Database [45]. The predicted results were named according to the Cytochrome P450 Database (https://cyped.biocatnet.de/, accessed on 5 June 2024). The e-value cutoff was set to be less than or equal to 1 × 10^−5^.

### 2.5. Comparative Genomics Analysis

The genome and protein sequences of other macrofungi were downloaded from the National Center for Biotechnology Information (https://www.ncbi.nlm.nih.gov/, accessed on 4 June 2024) (Appendix A). Carbohydrate-active enzymes (CAZymes) were predicted using dbCAN3 [46]. The online tool antiSMASH7.0 [47] was used to predict genes potentially related to secondary metabolites. OrthoFinder (v2.5.4) [48] software was used to construct the orthologous groups of 19 macrofungi. Single-copy orthologous genes were obtained through screening and used to construct the phylogenetic tree to reveal the phylogenetic relationships among *P. echinovolvatus* and other related taxa. The single-copy orthologs were further aligned using MUSCLE (v3.8.31) [49], and then the conserved region was extracted with the aid of Gblocks 0.91b [50] to obtain the codon-type nucleic acid comparison, which was concatenated for phylogenetic tree construction. Maximum likelihood (ML) estimation of the model parameters was performed using jModeltest v2.1.10 software [51] and GTR+I+G was the optimal model. PhyMLv3.0 [52] was used to construct the ML phylogenetic tree. Collinearity analysis was performed using TBtools-II [53]. The results were visualized using the ChiPlot Cloud Platform (https://www.chiplot.online/, accessed on 5 June 2024). The gene sequences of Hap1 and Hap2 were extracted from the Hap1 and Hap2 genome sequences and compared with each other using blastn (e-value < 1 × 10^−5^). Genes without aligned partners between Hap1 and Hap2 were considered unique to either Hap1 or Hap2.

## 3. Results and Discussions

### 3.1. Genome Sequencing and Assembly

We used the Illumina and PacBio sequencing platforms to sequence the genome of *P. echinovolvatus*. Illumina sequencing generated 7.04 Gb of clean data. Hi-C sequencing yielded 6.91 Gb of clean reads. Additionally, PacBio sequencing provided 24.37 Gb of HiFi reads (Appendix A).

The 7.04 Gb of clean data obtained through Illumina sequencing were used for the k-mer analysis to generate a histogram of the depth distribution of the sequencing data (k = 21) (Appendix A). The estimated genome size of *P. echinovolvatus* was 37.8 Mb, and the estimated heterozygosity was approximately 1.69%. Notably, two major peaks were observed in the graph, which were positioned between 50 and 150, and the first peak was higher than the second. This demonstrates that RITF7875 is a dikaryotic strain.

Based on the HiFi reads, we initially assembled two haploid genomes, Hap1 and Hap2, along with the reference genome, Monop. The Hap1, Hap2, and Monop genomes comprised 183, 122, and 76 contigs, and their sizes were 37.68 Mb, 38.46 Mb, and 43.85 Mb, respectively. The N50 was 0.65 Mb, 0.73 Mb, and 1.20 Mb for the Hap1, Hap2, and Monop genomes, and the guanine-cytosine (GC) content percentages were 43.96%, 43.97%, and 43.98%, respectively (Table 1). The contigs or scaffolds were clustered, sorted, and oriented using ALLHIC software to construct the chromosome-level genome, and interaction mapping was used to correct inconsistencies. Finally, three genomes with 11 chromosomes each were generated, with sizes of 35.46 Mb (Hap1), 36.11 Mb (Hap2), and 36.54 Mb (Monop), and scaffold N50 lengths of 3.31 Mb, 3.33 Mb, and 3.43 Mb, respectively (Table 1). Based on the interaction maps of the Hap1, Hap2, and Monop genomes, 11 distinct blocks were observed, which corresponded to 11 chromosomes. The strong correlation between the chromosome-level genome assembly and the Hi-C interaction maps indicates that this assembly was highly reliable (Appendix A).

Compared to the other two species in the *Phallus* genus, the genomes of the three species show significant differences. *P. indusiatus* has the largest genome size (67.32 Mb), while *P. rubrovolvatus* has the smallest (32.89 Mb). The proportion of the coding gene length to the total genome length is highest in *P. rubrovolvatus* (63.1%) and lowest in *P*. *echinovolvatus* (37.09%). The average coding gene length in *P. rubrovolvatus* is much greater than that in *P. echinovolvatus* and *P. indusiatus*, whereas *P. indusiatus* has a much higher gene count than *P. rubrovolvatus* and *P. echinovolvatus* (Table 1). These differences may be due to species-specific characteristics or variations in sequencing technologies; further investigation is needed to clarify the underlying causes.

Furthermore, the Benchmarking Universal Single-Copy Orthologs (BUSCOs, basidiomycota_odb11) were used to assess the assembly quality. A total of 1764 BUSCOs were identified in the genome assembly, and the completeness rates of the Hap1, Hap2 and Monop genomes were 95.6%, 95.8% and 96.7%, respectively (Table 2). These results indicate that the *P. echinovolvatus* genome sequence had high completeness and contiguity.

### 3.2. Genome Component of P. echinovolvatus

#### 3.2.1. Gene Prediction

In the reference genome (Monop), a total of 10,251 coding genes with an average length of 1582 bp were predicted. The cumulative length of these genes amounted to 16.21 Mb, accounting for 37.09% of the genome. The average numbers of exons and introns per gene were 7.51 and 6.51, respectively. The two haploid genomes predicted 9316 and 9328 coding genes, which represented 39.38% and 38.47% of the genome, respectively. The average counts of exons and introns per gene were 7.54 and 6.54, respectively (Appendix A).

#### 3.2.2. Repeat Sequences Prediction

Repetitive sequences are widespread in eukaryotic genomes and are considered ancient components of the genome. They can be divided into two categories: interspersed repeats (IRs) and tandem repeats (TRs) [54]. TRs include minisatellite DNA, microsatellite DNA, and satellite DNA. Discrete repeats include retrotransposons and DNA transposons [55]. Repeat sequences of *P. echinovolvatus* were identified using both homology-based and de novo strategies. Repeat sequences accounted for 17.73%, 14.45%, and 15.84% of the Monop, Hap1, and Hap2 genomes, respectively. Most IRs were long terminal repeats, long interspersed nuclear elements, and DNA transposable elements. Most TRs comprised minisatellite DNA and microsatellite DNA (Appendix A). Repetitive sequences play an important role in maintaining the spatial structure of chromosomes, gene expression, genetic regulation, and biological evolution. Accurately understanding the structure of the repetitive sequences in the genome contributes to genome research [56].

#### 3.2.3. Non-Coding RNA Prediction

The predictions of non-coding RNAs in the *P. echinovolvatus* genome are shown in Appendix A. A total of 107 tRNAs, five rRNAs, and 14 snRNAs were predicted in the reference genome (Monop). Among the tRNAs, 41 were unidentified anticodons, and the remaining anticodon tRNAs comprised 17 common amino acid codons. In the Hap1 and Hap2 genomes, 103 and 94 tRNAs, four and five rRNAs, and 12 and 14 snRNAs were predicted, respectively.

### 3.3. Annotation of Gene Function

Functional analysis of the predicted gene sequences was conducted using multiple public databases (NR, GO, KOG, KEGG, P450, Swiss-Prot, HMHMM, Pfam). This analysis resulted in the identification of 8851, 8106, and 8080 annotated genes for the Monop, Hap1, and Hap2 genomes, respectively (Appendix A). Based on the gene annotation results, we conducted further analysis of the reference genome (Monop).

#### 3.3.1. GO Annotation

A total of 3402 genes were annotated in the GO database. The GO terms in the Biological Process category were the most common and included “cellular process” (3195 genes), “metabolic process” (2516 genes), and “biological regulation” (1427 genes); within the Cellular Component category, the GO terms “cellular anatomical entity” (3191 genes), “intracellular” (3135 genes), and “protein-containing complex” (1613 genes) were detected; and within the Molecular Function category, the GO terms “catalytic activity” (1659 genes) and “binding” (1576 genes) were detected (Figure 2).

#### 3.3.2. KEGG Annotation

Annotations from the KEGG database were obtained for 3566 genes, which accounted for 34.79% of all the genes. These genes were classified into six major categories: Metabolism (11 branches, 1719 genes), Genetic Information Processing (4 branches, 925 genes), Environmental Information Processing (3 branches, 503 genes), Cellular Processes (5 branches, 814 genes), Organismal Systems (10 branches, 799 genes), and Human Diseases (12 branches, 1276 genes). Within the Metabolism category, the 1719 genes were further classified into 11 subcategories, primarily “Carbohydrate metabolism” (369 genes), “Amino acid metabolism” (346 genes), “Lipid metabolism” (236 genes), “Energy metabolism” (141 genes), “Xenobiotics biodegradation and metabolism” (143 genes), “Metabolism of cofactors and vitamins” (125 genes), and “Glycan biosynthesis and metabolism” (117 genes) (Figure 3).

#### 3.3.3. KOG Annotation

The annotations derived from the KOG database are shown in Figure 3. A total of 5657 genes were annotated in the KOG database, which accounted for 55.18% of all the genes. The results revealed that the genes were enriched in various metabolic processes, including “Posttranslational modification, protein turnover, chaperones” (547 genes), “Signal transduction mechanisms” (471 genes), “Secondary metabolite biosynthesis, transport and catabolism” (419 genes), “Intracellular trafficking, secretion, and vesicular transport” (353 genes), “Carbohydrate transport and metabolism” (308 genes), “Translation, ribosomal structure and biogenesis” (308 genes), “Amino acid transport and metabolism” (303 genes), and “Energy production and conversion” (301 genes) (Figure 4).

#### 3.3.4. The Cytochromes P450 (CYPs) Family

Cytochrome P450 enzymes (CYPs) are terminal oxidases and the major catalysts involved in the metabolism of drugs involved in detoxification, the degradation of xenobiotics, and the biosynthesis of secondary metabolites [57,58]. *P. echinovolvatus* had a total of 385 CYP genes in 30 families according to Nelson’s nomenclature [59]. The CYP51 family contained the greatest number of genes (116 genes), followed by the CYP53 family (42 genes), CYP620 family (28 genes), CYP504 family (28 genes), CYP512 family (23 genes), CYP125 family (22 genes), CYP505 family (20 genes), and CYP78 family (18 genes) (Appendix A). *P. echinovolvatus* had a few P450 genes involved in KEGG pathways. The results of the gene function annotation revealed that 10 and 11 genes were involved in “Metabolism of xenobiotics by cytochrome P450” and “Drug metabolism cytochrome P450”, respectively.

### 3.4. Comparative Genomics Analysis

#### 3.4.1. CAZymes

Complex carbohydrates are widely distributed in nature, and they play numerous biological roles within organisms, including serving as structural molecules, sources of energy, and mediators of cell recognition within the same organism or between different organisms. Carbohydrate-active enzymes are involved in the assembly and breakdown of these complex carbohydrates [60]. A total of 433 CAZyme genes were identified from the genome of *P. echinovolvatus*, and these were distributed across 103 CAZyme families, including 199 glycoside hydrolases (GHs, 49 families, 45.96%), 90 glycosyl transferases (GTs, 27 families, 20.79%), 13 polysaccharide lyases (PLs, 5 families, 3.00%), 25 carbohydrate esterases (CEs, 7 families, 5.77%), 100 auxiliary activity enzymes (AAs, 11 families, 23.09%), and six carbohydrate-binding modules (CBMs, 4 families, 1.39%) (Figure 5a and Appendix A).

Fungi play a key role in the degradation of plant biomass. Symbiotic fungi can obtain the nutrients needed for growth and development by establishing a symbiotic relationship with the host, and saprophytic fungi mainly obtain nutrients by decomposing humus. Under natural conditions, *P. echinovolvatus* usually grows in the decayed litter layer of bamboo forest. The number of genes encoding carbohydrate enzymes predicted in the genome of *P. echinovolvatus* was more than that in symbiotic fungi but less than that in most saprophytic fungi. The genome of *P. echinovolvatus* contained genes that encode enzymes required for cellulose degradation, such as GH1, GH3, GH5, GH6, GH7, GH12, GH74, GH92, and AA9, as well as hemicellulases, including GH10, GH11, GH16, GH30, GH115, CE1, and CE2; ligninases, such as AA1, AA2, AA3, AA7, and AA5; and pectinases, such as GH2, GH28, GH35, GH43, GH51, GH53, GH78, GH88, GH93, GH105, and CE8. Additionally, in the *P. echinovolvatus* genome, there were six genes related to CBMs, including four families; most were related to the CBM91 and CBM20 families. Many biomass-degrading fungi commonly employ CBMs for plant cell wall degradation [61]. In the case of *P. echinovolvatus*, CBMs form chimeric enzymes with CAZymes from several families to enhance the hydrolysis of carbohydrates, as well as insoluble substrates such as cellulose, chitin, and starch. We compared the numbers of enzymes in different families in *P. echinovolvatus* with those in other mushrooms and found that the numbers of AA9, GH5, GH43, CBM13, and CE4 were significantly lower in *P. echinovolvatus* than in most saprophytic fungi. Both *P. echinovolvatus* and *P. rubrovolvatus* lacked genes encoding PL1, PL3, and PL4 for pectin degradation. These results suggest that *P. echinovolvatus* might have relatively weak plant cell-wall-degrading abilities, which might contribute to its longer growth cycle (Figure 5 and Appendix A).

Fungi rely on CAZymes to degrade complex polysaccharides such as cellulose, hemicellulose, and lignin, which allows them to access and utilize diverse carbon sources present in their environment. The CAZyme repertoire of a fungus reflects its ability to colonize specific substrates. Interestingly, *P. indusiatus* has a significantly higher number of CAZyme genes compared to *P. echinovolvatus* and *P. rubrovolvatus*—2.10 and 2.53 times higher, respectively, highlighting the significant expansion of the CAZyme repertoire in *P. indusiatus* compared to the other two species. This variation in the CAZyme gene numbers appears to be associated with differences in the genome size, suggesting that genome expansion may play a role in the diversification of CAZyme repertoires among these species. The extensive CAZyme repertoire of *P. indusiatus* suggests an ecological adaptation to niches that demand a broader range of enzymatic activities, potentially targeting complex substrates like bamboo or other lignocellulosic materials. In contrast, *P. echinovolvatus* and *P. rubrovolvatus* may specialize in simpler substrates, necessitating fewer CAZymes. The larger number of CAZyme genes likely provides *P. indusiatus* with greater metabolic flexibility, allowing it to adapt to diverse or challenging environments. While *P. indusiatus* stands out for its extensive CAZyme gene set, the specific evolutionary and ecological drivers of this disparity remain unclear and warrant further investigation, particularly in the context of substrate availability and habitat specialization.

#### 3.4.2. Secondary Metabolites

During the growth of edible fungi, a wide variety of secondary metabolites are produced, and these serve as important sources of bioactive substances in edible fungi. These bioactive substances include amino acids, polysaccharides, terpenes, vitamins, and other active compounds. The secondary metabolites often exhibit unique biological activities. For example, they have various functions, such as reducing blood lipid levels, inhibiting tumor growth, enhancing immune responses, and regulating metabolism. Consequently, these secondary metabolites have significant implications for modern pharmaceutical development [62,63]. The gene clusters involved in the biosynthesis of secondary metabolites in the *P. echinovolvatus* genome were predicted using antiSMASH. As shown in Figure 6a and Appendix A, a total of 42 predicted secondary metabolite gene clusters were identified, including 32 terpene gene clusters, two T1PKS gene clusters, two siderophore gene clusters, one indole gene cluster, and five NRPS gene clusters. T1PKSs are involved in the biosynthesis of complex polyketides, which include many clinically important compounds, such as antibiotics, antifungals, and anticancer agents [64]. The relatively low number of T1PKS clusters in *P. echinovolvatus* compared to other fungi suggests limited production of complex polyketides, but the presence of these clusters indicates potential for unique bioactive compounds. Similarly, indole clusters are sparse, with most species having zero to four clusters. *P. echinovolvatus* has one cluster, indicating limited indole metabolite biosynthesis compared to other fungi. Siderophores are iron-chelating compounds that facilitate iron uptake in iron-limited environments. Siderophore clusters are fairly evenly distributed, ranging from zero to five clusters [65]. *P. echinovolvatus* has two clusters, indicating its potential for iron-chelating activities, which are vital for fungal growth and survival. Additionally, non-ribosomal peptide synthetases (NRPSs) are involved in the biosynthesis of peptides with various biological activities, including antibiotics, immunosuppressants, and siderophores [66,67]. The number of NRPS gene clusters is often closely related to the ecological adaptability and diversity of secondary metabolites. A total of five NRPS gene clusters were predicted in the genome of *P. echinovolvatus*, indicating a lower count compared to certain other fungi. This suggests that *P. echinovolvatus* may rely more on other secondary metabolic pathways, such as terpenoid biosynthesis, for environmental adaptation.

It is worth noting that terpene-related gene clusters account for the largest proportion of these clusters, which is lower than that of *P. indusiatus* (46 clusters) but significantly higher than in other macrofungi, indicating a much greater potential for synthesizing terpene compounds compared to other macrofungi. Mapping the terpene gene clusters to the chromosomes revealed their distribution across all the chromosomes (Figure 6b and Appendix A).

Terpenoid compounds are a class of natural organic compounds formed by the polymerization of isoprene units (C5H8). They are important components of secondary metabolites, widely found in microorganisms. They have important physiological activities and biological functions, and they are particularly important in drug development and growth regulation. For example, guanacastane-type diterpenoids with antitumor activity have been isolated from *Coprinus radians* [68]. The triterpene acid extract from *Ganoderma lucidum* can inhibit the activity of human hepatoma SMMC-7721 and human colon carcinoma SW620 cells [69]. Terpenoid compounds perform various crucial functions within organisms. They act as volatile substances to deter pests and pathogens, attract pollinators or seed dispersers, and help microorganisms adapt to environmental stresses [70,71,72]. Previous studies have indicated that one of the primary constituents of the volatile aroma of bamboo fungi is volatile terpenoids, such as geranylacetone, β-patchoulene, limonene, alpha-chamigrene, cedrene, γ-selinene, and cedrol [73,74]. Plants utilize scent to attract animals to aid in seed dispersal and reproduction. Possibly similar to the reproductive strategy of plants, when *P. echinovolvatus* matures, the surface of the cap secretes mucilage containing numerous sexual spores and emits a strong odor, which might attract more insects to facilitate spore dissemination. The genomes of both *P. indusiatus* and *P. echinovolvatus* are significantly enriched with gene clusters associated with terpene biosynthesis; this likely stems from long-term adaptation to the environment. Isopentenyl pyrophosphate (IPP) and dimethylallyl pyrophosphate (DMAPP) are common precursors for the biosynthesis of terpenes. IPP and DMAPP are primarily derived from the mevalonate (MVA) and methylerythritol phosphate (MEP) pathways [75,76]. Based on the KEGG annotation results, there were 15 enzymes involved in “terpenoid backbone biosynthesis (map00900)” in *P. echinovolvatus*, where acetyl-CoA C-acetyltransferase, isopentenyl-diphosphate delta-isomerase, and phosphomevalonate kinase were encoded by two genes each, and the remaining 12 enzymes were encoded by a single gene. Similar to other basidiomycetes, the MVA pathway was detected in *P. echinovolvatus*; however, the MEP/DOXP pathway was not detected (Appendix A). There were two enzymes involved in “Sesquiterpenoid and triterpenoid biosynthesis (map00909)”, where farnesyl-diphosphate farnesyltransferase was encoded by three genes and squalene monooxygenase was encoded by one gene. Furthermore, we identified two genes encoding lanosterol synthase (LSS) (AMY2.Monop07530.1, AMY2.Monop07562.1 [EC:5.4.99.7]); one enzyme was involved in “Diterpenoid biosynthesis(map00904)”, six enzymes were involved in “Ubiquinone and other terpenoid-quinone biosynthesis (map00130)”, NAD(P)H dehydrogenase (quinone) was encoded by four genes, and the remaining five enzymes were encoded by a single gene. However, no enzymes associated with the “monoterpenoid biosynthesis (map00902)” pathway were identified.

The diversity of the secondary metabolite gene clusters underscores the variability in the biosynthesis of secondary metabolites across different species. A comparison with other macrofungi revealed that *P. echinovolvatus* exhibits significant potential for terpene biosynthesis, making it a promising source of diverse terpenoids that are not only ecologically significant but also possess substantial biological and pharmaceutical value. Its rich terpenoid diversity positions it as a valuable candidate for natural product discovery, particularly in drug development and other biotechnological applications. To fully realize this potential, future studies should focus on elucidating the terpene biosynthetic pathways of *P. echinovolvatus* through transcriptomic, proteomic, and metabolomic, combined with bioinformatics analysis and experimental validation. This approach will enable a deeper understanding of the specific functions of these gene clusters and exploration of their application potential.

#### 3.4.3. Phylogenetic Analysis

The evolutionary relationships between *P. echinovolvatus* and 18 other fungi (*T. melanosporum* from Ascomycota as the outgroup) were investigated. A total of 629 single-copy orthologous gene families were found and used to construct the phylogenetic tree, and 844 genes were unique to *P. echinovolvatus* (Appendix A). As shown in Figure 7, *P. echinovolvatus* was nested within a large clade formed by five species under Phallales and clustered with *C. columnatus*, indicating that *P. echinovolvatus* is closely related to *C. columnatus*. *P. echinovolvatus* and *C. columnatus* both belong to the family Phallaceae, and the results of the phylogenetic analysis were consistent with the classification results based on the morphological and molecular traits.

#### 3.4.4. Collinearity Analysis

Based on the phylogenetic analysis, we conducted a genome collinearity analysis using *C. columnatus*, which is closely related to *P. echinovolvatus* (its genome comprises 11 long contigs and eight short contigs), and *P. ostreatus*, which is more distantly related (its genome comprises 11 chromosomes). We observed pronounced collinearity between *P. echinovolvatus* and *C*. *columnatus*, and the collinearity between *P. echinovolvatus* and *P. ostreatus* was not significant. The results indicated the high reliability of the *P. echinovolvatus* genome assembly. Previous studies have found that structural genomic variants play an important role in the evolution of species [77]. According to the results of the collinearity analysis, genomic structural variations were observed both within and between chromosomes. Chromosomal segment heterotopy was observed in chromosomes 5, 7, 9, and 10 in *P. echinovolvatus* and *C. columnatus*. Major chromosomal rupture and fusion events occurred between chromosomes 2, 3, and 11. Obvious rearrangements, ruptures, and fusion events were detected in chromosome 1 (Figure 8).

#### 3.4.5. Comparative Analysis of Hap1 and Hap2

The length and gene count of each chromosome in Hap1 and Hap2 are shown in Table 3. A total of 156 pairs of one-to-one matching gene blocks were detected, which accounted for 84.62% of all the genes. Strong collinearity was detected between Hap1 and Hap2; chromosomes 2, 3, 5, 6, 7, 9, and 10 in Hap1 corresponded to chromosomes 3, 7, 6, 2, 5, 10, and 9 in Hap2, respectively. Chromosomes 5, 6, and 9 in Hap1 and chromosomes 2 and 5 in Hap2 experienced minor fragmentation and fusion events. These results suggested that the two haploid nuclei might have been derived from different parents (Figure 9).

Subsequently, non-collinear gene sequences were extracted from the genome and subjected to blastn alignment for the two haplotypes (e-value < 1 × 10^−5^). There were 256 unique genes in Hap1 and 370 unique genes in Hap2. These unique genes were widely distributed across each chromosome, with a predominant location on chromosome 2 in Hap1 and on chromosomes 5 and 8 in Hap2 (Figure 10).

Functional enrichment analysis was conducted for unique genes, and the KEGG annotation results indicated that the unique genes in Hap1 were primarily enriched in pathways related to “Biosynthesis of secondary metabolites”, “Metabolic pathways”, and “Ubiquitin-mediated proteolysis”. The unique genes in Hap2 were significantly enriched in “Metabolic pathways” and “Oxidative phosphorylation” (Appendix A). The results of the GO analysis indicated that the unique genes in Hap1 were mainly enriched in processes related to “cellular protein modification process”, “macromolecule modification”, “organelle membrane”, and “protein modification process”. The unique genes in Hap2 were primarily enriched in processes related to “envelope”, “organelle envelope”, and “oxidoreductase activity” (Appendix A).

The genome of the dikaryotic strain of *P. echinovolvatus* is highly heterozygous, and the results from the phased assembly and annotation indicate genetic differences between the two nuclei. Given the limitations of current sequencing technologies and analytical tools, future work involving the mononucleation of dikaryotic mycelium, followed by separate sequencing of the resulting mononuclear strains, could yield more accurate genomic information. This approach would also allow for a deeper investigation into the roles of each nucleus in the growth and reproduction of *P. echinovolvatus*.

## 4. Conclusions

In this study, we present the genomic information about *P. echinovolvatus* obtained by integrating different sequencing technologies. Functional annotations of the genomes were obtained using multiple public databases. This whole-genome assembly, along with the associated annotation data, represents the first chromosome-level genome assembly for *P. echinovolvatus*. These new data will aid future studies on the evolution of species and phylogenetic analyses based on genomic data. Furthermore, these data will be useful for breeding programs, as well as for studies of developmental mechanisms and the pathways underlying the biosynthesis of bioactive compounds in *P. echinovolvatus*. Our findings provide genetic and molecular insights into the evolutionary history of *P. echinovolvatus* and offer genomic resources to further facilitate gene editing to enhance desirable traits.

## Figures and Tables

**Figure 1 jof-11-00062-f001:**
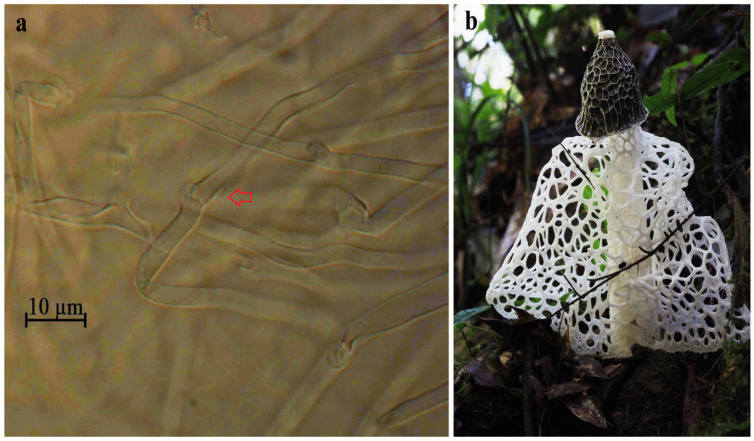
Dikaryotic mycelia and fruiting body of *P. echinovolvatus*. (**a**) Microscopic observation of dikaryotic mycelia of the RITF7875 strain where the clamp connections are indicated by arrows. (**b**) Cultivated *P. echinovolvatus* under trees in Guangdong Province, China.

**Figure 2 jof-11-00062-f002:**
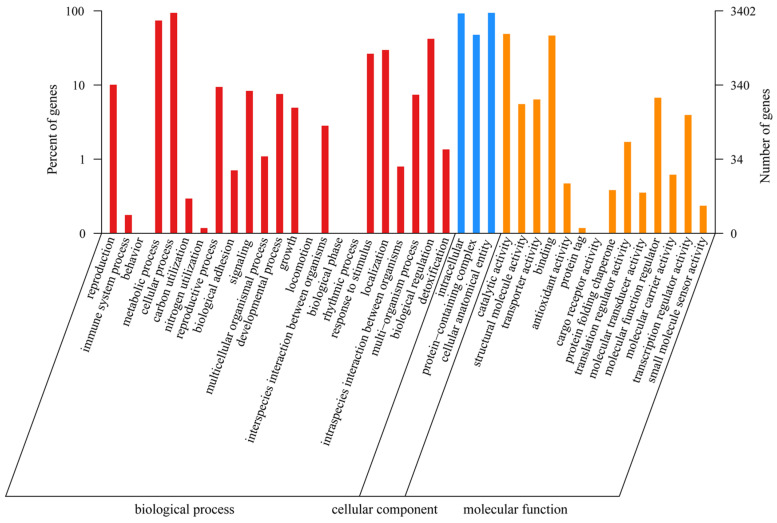
The GO function annotation of *P. echinovolvatus*.

**Figure 3 jof-11-00062-f003:**
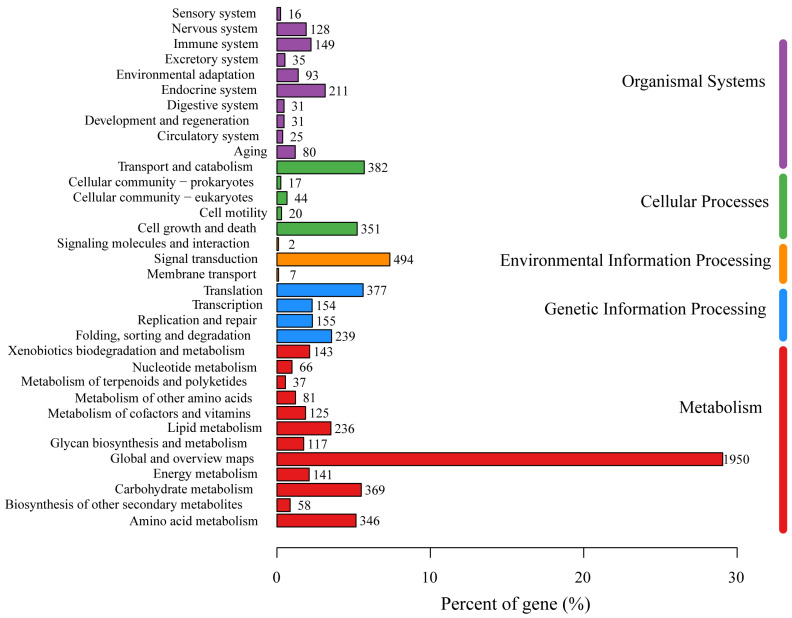
The KEGG function annotation of *P. echinovolvatus*.

**Figure 4 jof-11-00062-f004:**
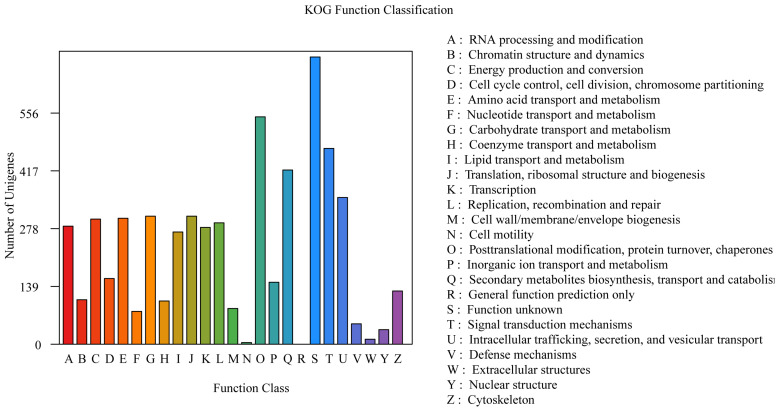
The KOG function annotation of *P. echinovolvatus*.

**Figure 5 jof-11-00062-f005:**
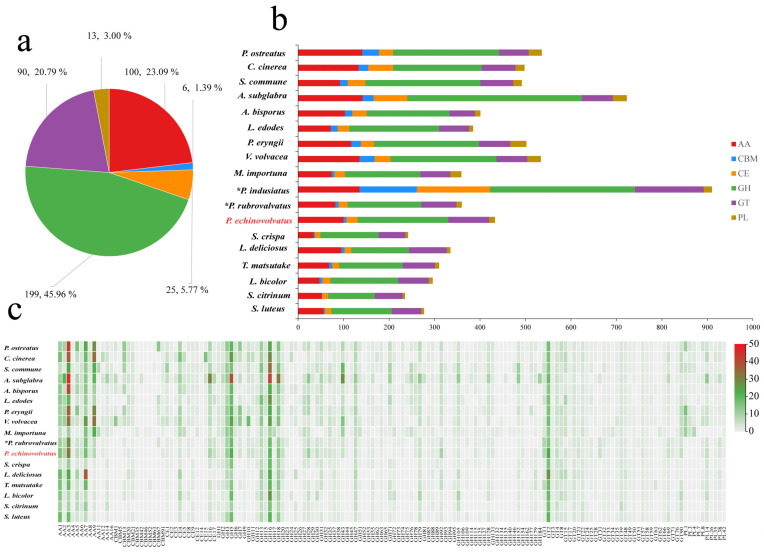
CAZymes in *P. echinovolvatus* and other fungi. (**a**) The distribution of CAZymes in *P. echinovolvatus*. (**b**) The distribution of CAZymes in another 23 fungi. (**c**) Heatmap representing the CAZyme families distributed in *P. echinovolvatus* and other fungi. GH, glycoside hydrolase; GT, glycosyltransferase; PL, polysaccharide lyase; CE, carbohydrate esterase; CBM, carbohydrate-binding module; AA, auxiliary activity. * Referenced [20,21]. The red font is the target species sequenced in this paper.

**Figure 6 jof-11-00062-f006:**
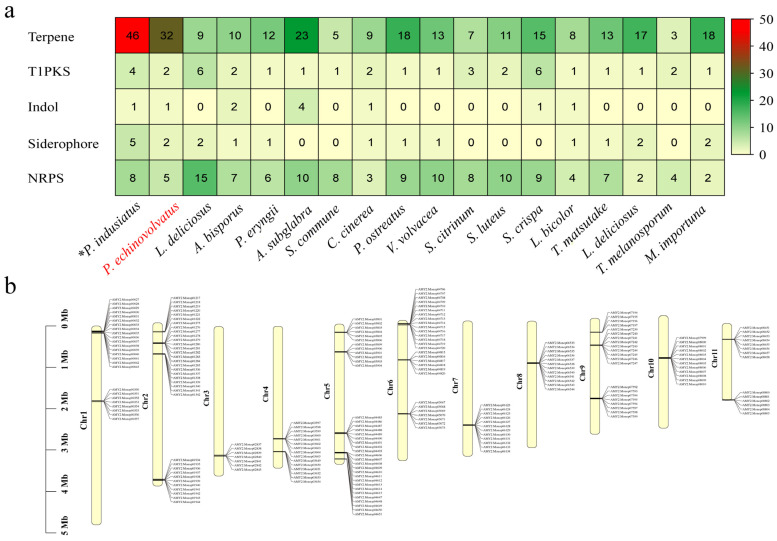
Secondary metabolite prediction results for *P. echinovolvatus*. (**a**) Comparison of the secondary metabolites of *P. echinovolvatus* with those of other fungi. (**b**) The distribution of gene clusters encoding sesquiterpenes on chromosomes. NRPS, non-ribosomal peptides; PKS, polisyketides. * Referenced [20]. The red font is the target species sequenced in this paper.

**Figure 7 jof-11-00062-f007:**
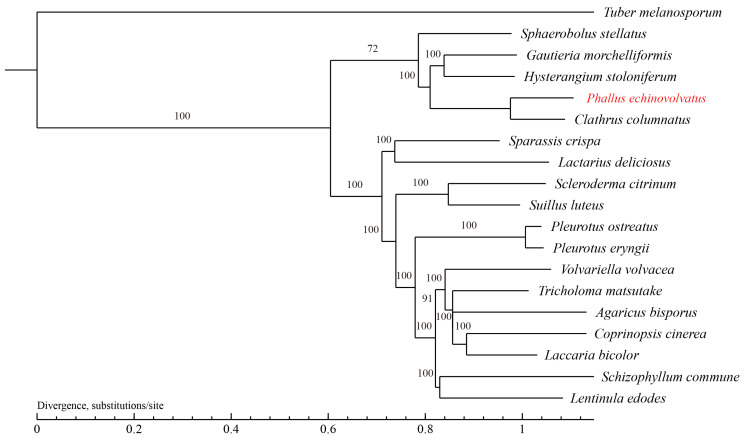
Phylogenetic tree of 19 fungi based on single-copy orthologous genes. *T. melanosporum* served as the outgroup. The red font is the target species sequenced in this paper.

**Figure 8 jof-11-00062-f008:**
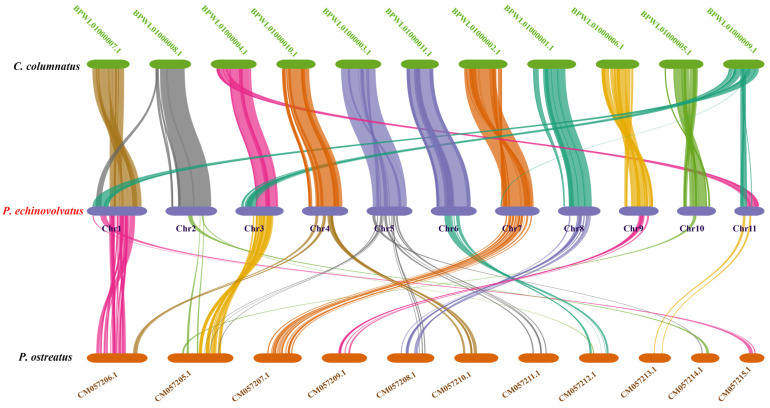
The genome collinearity among *P. echinovolvatus*, *C. columnatus*, and *P. ostreatus.* The red font is the target species sequenced in this paper.

**Figure 9 jof-11-00062-f009:**
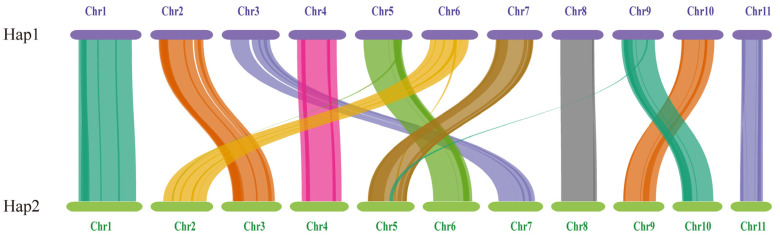
The genome collinearity among Hap1 and Hap2.

**Figure 10 jof-11-00062-f010:**
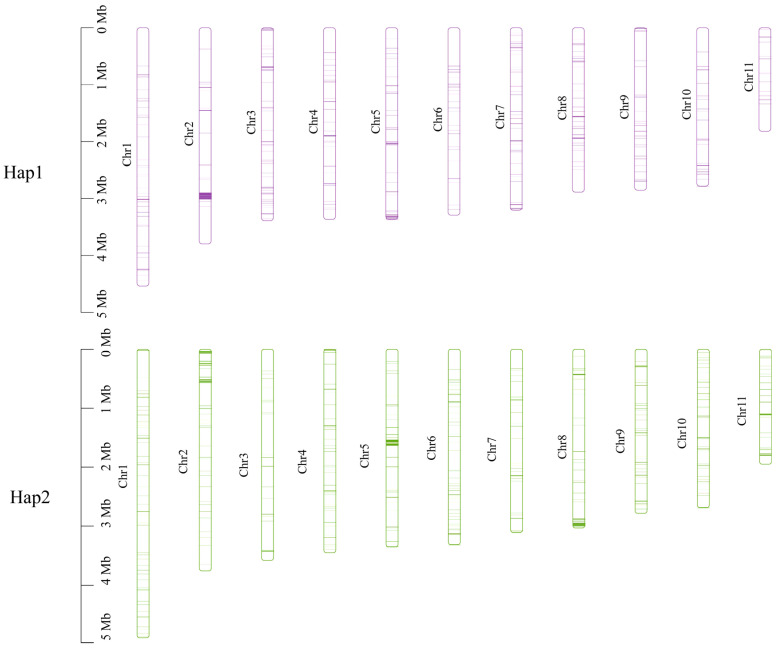
Location of the unique genes found within each of the two monokaryons on each chromosome.

**Table 1 jof-11-00062-t001:** Genome assembly features of *P. echinovolvatus*, *P. rubrovolvatus* and *P. indusiatus*.

Contents	Monop	Hap1	Hap2	*P. rubrovolvatus*	*P. indusiatus*
Sequencing Technology	Illumina + Hi-C + PacBio HiFi				Illumine + ONT
Number of Scaffolds	76	183	122	132	216
Total Length (Mb)	43.85	37.68	38.46	32.89	67.32
Scaffold N50 (Mp)	1.20	0.65	0.73	2.7	0.79
Guanine-Cytosine Content (%)	43.98	43.96	43.97	45.16	44.05
Gene Number	10,251	9316	9328	9725	19,909
Gene Total Length (Mb)	16.21	14.83	14.78	20.76	31.87
Gene Average Length (bp)	1582	1592	1585	2135.07	1601
Gene Length/Genome (%)	37.09	39.38	38.47	63.1	47.25
Pseudo-Chromosome	11	11	11	11	_
Pseudo-Chromosome Total Length (Mb)	36.54	36.11	35.46	28.24	_
Pseudo-Chromosome N50 Length (Mb)	3.42	3.38	3.36	2.75	_
Chromosome Anchoring Rate for Contigs (%)	83.58	94.16	93.98	85.57	_

**Table 2 jof-11-00062-t002:** Statistics from the BUSCO analysis of the *P. echinovolvatus* genome.

Term	Hap1		Hap2		Monop	
	BUSCO Number	Percentage	BUSCO Number	Percentage	BUSCO Number	Percentage
Complete BUSCOs	1687	95.6	1691	95.8	1706	96.7
Complete and single-copy BUSCOs	1553	88	1579	89.5	1460	82.8
Complete and duplicated BUSCOs	134	7.6	112	6.3	246	13.9
Fragmented BUSCOs	13	0.7	12	0.7	13	0.7
Missing BUSCOs	64	3.7	61	3.5	45	2.6
Total BUSCO groups searched	1764	1.0	1764	1.0	1764	1.0

**Table 3 jof-11-00062-t003:** The length and gene count of each chromosome.

ID	Hap1		Hap2	
	Length (bp)	Gene Number	Length (bp)	Gene Number
Chr1	4,557,232	1177	4,928,922	1169
Chr2	3,821,430	812	3,777,475	980
Chr3	3,400,287	844	3,601,952	766
Chr4	3,382,853	923	3,473,486	949
Chr5	3,377,001	1052	3,363,986	614
Chr6	3,305,313	867	3,326,919	1032
Chr7	3,217,999	586	3,117,068	719
Chr8	2,906,280	763	3,049,163	783
Chr9	2,867,821	579	2,808,455	679
Chr10	2,792,080	687	2,697,050	574
Chr11	1,831,909	454	1,961,784	465
Total	35,460,205	8744	36,106,260	8730

## Data Availability

The genome sequencing of *P. echinovolvatus* RITF7875 generated for this study has been submitted to the NCBI (BioProject: PRJNA 1192246).

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
