# Peer review of "Chromosome-Level Genome Assembly and Annotation of the Highly Heterozygous Phallus echinovolvatus Provide New Insights into Its Genetics"

_jof, 2025, doi:10.3390/jof11010062_

Round 1
Reviewer 1 Report
Dear authors! Thank you for the work provided. The article is very interesting, full of information. It is devoted to the genetic study of the fungus Dictyophora echinovolvata. The authors use modern methods of molecular biology and bioinformatics in their work. The relevance of this article leaves no doubt. Figure 8 is the decoration of this article. There are a number of comments that I have indicated above. I believe that after a small revision of the manuscript, the article can be published in the journal "JoF". December 11, 2024 Respectfully Yours, reviewer
The authors need to work on the design of the "References" according to the MDPI rules.
Author Response
1. Summary
Dear reviewer,
Thank you for your letter and for the reviewers’comments concerning our manuscript entitled “Chromosome-level Genome Assembly and Annotation of the Highly Heterozygous Phallus echinovolvatus Provide new In-sights into its Genetics” (ID: jof-3374034). Those comments are all valuable and very helpful for revising and improving our paper, as well as the important guiding significance to our researches. We have studied comments carefully and have made correction which we hope meet with approval. Revised portion are marked in red in the paper. The main corrections in the paper and the responds to the reviewer’s comments are as flowing:
Comments 1: [In my opinion, the information about secondary metabolites can be removed from the title, since only terpenoids were studied in the work.]
Response 1: Thank you for pointing this out. We agree with your suggestion and have revised the title of the manuscript accordingly to better reflect the content of the study. The new title is:
"Chromosome-level Genome Assembly and Annotation of the Highly Heterozygous Phallus echinovolvatus Provide New Insights into Its Genetics."This change has been updated on the title page and throughout the manuscript where applicable (Line 1–3). We believe this revised title more accurately represents the scope and findings of our research.
Comments 2: [The Introduction does not contain enough information about the global distribution of the studied fungus and its use in the world, in countries other than China]
Response 2: Thank you for your valuable suggestion. We have revised the Introduction to include more detailed information about the global distribution and uses of the studied fungus in countries beyond China. Additionally, in accordance with the latest taxonomic updates, we have updated the Latin name of the species from Dictyophora echinovolvata to Phallus echinovolvatus throughout the manuscript. Line 28–44
Comments 3: [At the end of the Introduction paragraph, the existing results of the study are written, but it was necessary to write the purpose of the study.]
Response 3: Thank you for your constructive comment. We have revised the end of the Introduction to clearly state the purpose of the study. The revised text now explicitly outlines the objectives and significance of our research, emphasizing the contributions this study aims to make. Lines 75–84 .
Comments 4: [The secondary metabolites in the "Results" section are only terpenoid compounds. However, terpenoid compounds are very diverse and include many other substances that are not discussed in Section 3.4.2.]
Response 4: Thank you for highlighting this important point. We have revised the discussion in Section 3.4.2 . Additionally, we have expanded the Results section to describe other secondary metabolite gene clusters identified in the genome, providing a more comprehensive overview of the secondary metabolism potential of Phallus echinovolvatus. Lines 371–392 and Lines 441–452.
Comments 5: [ Section 3 is titled "Results" but there is no "Discussion" section. Therefore, a new "Discussion" section should be added to the article or Section 3 should be renamed "Results and Discussion" and more discussion added.]
Response 5: Thank you for this valuable suggestion. We have addressed this concern by renaming Section 3 to "Results and Discussion" The additional discussion includes an interpretation of the results, linking them to existing literature and emphasizing their broader implications.We believe these changes improve the manuscript's structure and ensure a more comprehensive presentation of our findings and their significance.Lines 371–392 、Lines 441–452 and line 345–362.
Comments 6: [Unfortunately, I did not find in the file "Additional file" table S2 (line 160), table S1 (line 164), table S3 (line 206), table S6, which is referred to in the article for term 230. In the file "Additional file" there are only figures. In addition, the references in the text of the articles in the tables S are not in order.]
Response 6: Thank you for bringing this to our attention. We have carefully reviewed and addressed the issues related to the additional tables and their references:
Availability of Tables: All supplementary tables (S1–S13) have now been clearly organized and marked in red within the manuscript for easier identification.
Due to the large size and number of tables (13 in total), we have separated the supplementary materials into two files:
Excel File: Contains all supplementary tables (Tables S1–S13).
Word Document: Contains supplementary figures.
Both files have been packaged and uploaded as supplementary materials to ensure accessibility.
Comments 7: [In figure 5, the color chart is poorly distinguishable, especially when connected to the scale. It is necessary to make the figure more readable.]
Response 7: Thank you for highlighting this issue. We have redrawn Figure 5 to improve its readability, including adjustments to the color scheme and scale to ensure the data is clearly distinguishable. The updated figure has been incorporated into the revised manuscript.
Additionally, the data used for creating Figures 5 and 6 have been provided in the supplementary tables (Table S8 and Table S9) to allow for a more comprehensive understanding of the figures. These tables are included in the supplementary Excel file uploaded alongside the manuscript.
Comments 8: [The authors need to work on the design of the "References" according to the MDPI rules.]
Response 8: Thank you for pointing this out. We have revised the references section to fully comply with the MDPI formatting guidelines.
2. Additional clarifications
We would like to clarify that the Latin name of the studied organism has been updated from Dictyophora echinovolvata to Phallus echinovolvatus in accordance with the latest taxonomic revisions. This change has been consistently applied throughout the manuscript, including the title, main text, figures, tables, and supplementary materials.We believe this update ensures the scientific accuracy and consistency of the manuscript.
Reviewer 2 Report
The use of languages in many places in the manuscript doesn’t provide a reader without genome expertise enough context. Certain terminologies are indeed well-known in molecular biology/genetics but if a researcher from an orthogonal field reads the manuscript, it can feel less intuitive. One of such examples is the use of acronyms without explaining them at their first occurrence.
Another suggestion is to format the figures cleverly so that the appropriate amount and detail of information is presented.
The Results section should be renamed to Results and Discussion.
Line 28 – some names are included on this line
Line 44 – if Figure 1b is introduced first, swap 1a and 1b in the figure. A reference is needed here.
Line 47 – explain acronyms at their first occurrence, applicable for the whole manuscript.
Line 72 – “quick-frozen” to “flash-frozen”
Figure 1 – see comment for Line 44. Also reformat the caption so that it stays with the figure and is not separated to another page.
Line 79 – all suppliers’ name and location should be noted so that readers can replicate the experiment reliably.
Table 1- spell out “GC” in GC-Content
Figure 5 – in 5c the axes are not legible. Please include a larger figure in the SI. The same comment applies to figure 6b.
Author Response
1. Summary
Dear reviewer,
Thank you for your letter and for the reviewers’ comments concerning our manuscript entitled “Chromosome-level Genome Assembly and Annotation of the Highly Heterozygous Phallus echinovolvatus Provide new In-sights into its Genetics” (ID: jof-3374034). Those comments are all valuable and very helpful for revising and improving our paper, as well as the important guiding significance to our researches. We have studied comments carefully and have made correction which we hope meet with approval. Revised portion are marked in red in the paper. The main corrections in the paper and the responds to the reviewer’s comments are as flowing:
Comments 1: [The introduction failed to elaborate the goal of the study.]
Response 1: Thank you for your valuable suggestion. We have revised the Introduction to include more detailed information about the global distribution and uses of the studied fungus in countries beyond China. Additionally, in accordance with the latest taxonomic updates, we have updated the Latin name of the species from Dictyophora echinovolvata to Phallus echinovolvatus throughout the manuscript. Line 28–44 and Line 75–84.
Comments 2: [Materials and equipment used were missing details such as the location of certain suppliers.]
Response 2: Thank you for highlighting this important detail. We have revised the manuscript to include the missing information about the materials and equipment used, including the location of the suppliers. These updates can be found on Lines 98–99 of the revised manuscript.The updated text now specifies the supplier names and locations, ensuring the methods section provides sufficient detail for reproducibility.
Comments 3: [Line 28–some names are included on this line, Line 47-explain acronyms at their first occurrence, applicable for the whole manuscript and Table 1–spell out “GC” in GC–Content.]
Response 3: Thank you for pointing this out. We have made the following changes to address this comment: Throughout the manuscript, all acronyms have been explained at their first occurrence to ensure clarity for readers unfamiliar with the terms.
Comments 4: [The Results section should be renamed to Results and Discussion.]
Response 4: Thank you for this valuable suggestion. We have addressed this concern by renaming Section 3 to "Results and Discussion" The additional discussion includes an interpretation of the results, linking them to existing literature and emphasizing their broader implications.We believe these changes improve the manuscript's structure and ensure a more comprehensive presentation of our findings and their significance.Lines 371–392 、Lines 441–452 and line 345–362.
Comments 5: [Line 72–“quick-frozen” to “flash-frozen”. ]
Response 5: Thank you for your suggestion. We have revised the text accordingly, changing "quick-frozen" to "flash-frozen" on Line 72 in the manuscript. We appreciate your attention to detail and believe this change enhances the scientific accuracy of the description.
Comments 6: [“Line 44–if Figure 1b is introduced first, swap 1a and 1b in the figure. A reference is needed here”, “Figure 1–see comment for Line 44. Also reformat the caption so that it stays with the figure and is not separated to another page” and “Figure 5–in 5c the axes are not legible. Please include a larger figure in the SI. The same comment applies to figure 6b.” ]
Response 6: Thank you for highlighting this issue. We have redrawn Figure 5 to improve its readability, including adjustments to the color scheme and scale to ensure the data is clearly distinguishable. The updated figure has been incorporated into the revised manuscript.
Additionally, the data used for creating Figures 5 and 6 have been provided in the supplementary tables (Table S8 and Table S9) to allow for a more comprehensive understanding of the figures. These tables are included in the supplementary Excel file uploaded alongside the manuscript.
2. Additional clarifications
We would like to clarify that the Latin name of the studied organism has been updated from Dictyophora echinovolvata to Phallus echinovolvatus in accordance with the latest taxonomic revisions. This change has been consistently applied throughout the manuscript, including the title, main text, figures, tables, and supplementary materials.We believe this update ensures the scientific accuracy and consistency of the manuscript.